# Molecules Inducing Dental Stem Cells Differentiation and Bone Regeneration: State of the Art

**DOI:** 10.3390/ijms24129897

**Published:** 2023-06-08

**Authors:** Anastasia Ariano, Francesca Posa, Giuseppina Storlino, Giorgio Mori

**Affiliations:** Department of Clinical and Experimental Medicine, University of Foggia, Viale Pinto 1, 71122 Foggia, Italy; anastasia.ariano@unifg.it (A.A.);

**Keywords:** dental stem cells, mesenchymal stem cells, dental pulp stem cells, dental bud stem cells, bone regeneration, osteogenic differentiation

## Abstract

Teeth include mesenchymal stem cells (MSCs), which are multipotent cells that promote tooth growth and repair. Dental tissues, specifically the dental pulp and the dental bud, constitute a relevant source of multipotent stem cells, known as dental-derived stem cells (d-DSCs): dental pulp stem cells (DPSCs) and dental bud stem cells (DBSCs). Cell treatment with bone-associated factors and stimulation with small molecule compounds are, among the available methods, the ones who show excellent advantages promoting stem cell differentiation and osteogenesis. Recently, attention has been paid to studies on natural and non-natural compounds. Many fruits, vegetables, and some drugs contain molecules that can enhance MSC osteogenic differentiation and therefore bone formation. The purpose of this review is to examine research work over the past 10 years that has investigated two different types of MSCs from dental tissues that are attractive targets for bone tissue engineering: DPSCs and DBSCs. The reconstruction of bone defects, in fact, is still a challenge and therefore more research is needed; the articles reviewed are meant to identify compounds useful to stimulate d-DSC proliferation and osteogenic differentiation. We only consider the results of the research which is encouraging, assuming that the mentioned compounds are of some importance for bone regeneration.

## 1. Introduction

### 1.1. Dental Stem Cells (DSCs)

In modern medicine, mesenchymal stem cells (MSCs) are widely used as a therapeutic remedy: in fact, these multipotent cells are able to self-regenerate and differentiate towards cells of the mesodermal lineage, i.e., bone, cartilage, and fat cells [1]. MSCs, alone or integrated with scaffolds, can be used in the treatment of various pathologies. The efficacy of these cells has been evaluated in wound therapy [2], cartilage and bone dehiscence [3], graft rejection disease (GVHD) [4], cardiovascular [5], and neural disorders [6]. The most-studied MSCs are those resident in the bone marrow. However, bone-marrow harvesting implies physical pain and is psychologically demanding for the donor, so other sources of MSCs, such as dental tissues, have been investigated to overcome these complications. MSCs can derive from both mature and immature dental tissues [7]. Dental pulp stem cells (DPSCs) are usually extracted from the pulp of the wisdom tooth, while dental bud stem cells (DBSCs) come from the immature and therefore not-yet-calcified form of the tooth, i.e., the dental bud (DB) [8] (Figure 1). DBSCs and DPSCs express Nanog, OCT4, Sox2, c-Myc, and Klf4, transcription factors responsible for the pluripotency of stem cells, as they regulate multiple processes such as self-renewal, the expression of specific genes responsible for the differentiation in diverse tissues, and the transformation of stem cells into mature cells [8]. These stem cells have been seen as an attractive way to repair and regenerate damaged tissue, not only among dentists and orthopedists but also in the regenerative medicine community; they are a valid alternative solution to be used for bone regeneration thanks to their biological characteristics. Research has definitively demonstrated the differentiation of both bone and cartilage from MSCs [1,2,4]. However, the reconstruction of these hard tissues is still a challenge for regenerative medicine because, to date, treatments are not yet fully effective and are not always feasible.

### 1.2. Natural and Non-Natural Compounds

In this review, we present the study of molecules, of natural and non-natural compounds, and their effects on the osteogenic differentiation of cells from dental bud and dental pulp (Table 1).

Natural compounds are molecules that normally occur in nature, while non-natural compounds are chemicals that are synthesized and are not typically found in nature. Among the natural substances currently studied there are polyphenols which can be isolated from foods such as fruit, vegetables, wine, tea, etc. [9,10]. One of the main properties of these molecules is the antioxidant one which protects cells from damage caused by free radicals. Other natural compounds, such as vitamins, play essential roles in the development and functioning of the body; they contribute to the proper metabolism of nutrients, to the maintenance of the immune system [10], and to the prevention of a variety of pathological conditions. Last but not least, they can be obtained by diet or dietary supplements [11]. Many of these natural molecules can help our body deal with diseases and ailments, both preventively and curatively [12].

Studies have shown that some natural molecules can act as nutritional additives that improve mineralization and collagen formation; they open pathways for cell signaling and differentiation, thus promoting the development of new and stronger bone tissue.

In addition, non-natural molecules have shown similar beneficial effects on cell differentiation of bone tissues [13]. Furthermore, the combined use of stem cell therapy and such natural or pharmacological sources can enhance the osteogenic differentiation capacity of MSCs.

This collection of studies may provide the right input to continue the search for new molecules to be used for the bioengineering of bone.

**Table 1 ijms-24-09897-t001:** A schematic overview about effects of natural and non-natural compounds on DPSCs and DBSCs ( 

 increase, 
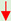
 decrease, 
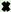
 absence).

DPSCs	DBSCs
Study	Natural Compounds	Effects	Study	Natural Compounds	Effects
Zhang et al., 2022 [14]Feng et al., 2016 [15]	Resveratrol	 osteogenic differentiation  antiox  Runx-2, BMP2, Col I, OCN, ALP	Di Benedetto et al., 2018 [16]	Polydatin	 osteogenic differentiation  mineral matrix deposition  ATF-4, OPN, ALP
	Flavonoids:		Di Benedetto et al., 2018 [16]	Resveratrol	 osteogenic differentiation  OPN, ALP
Fu et al., 2021 [17]	Taxifolin	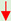 apoptosis  proliferation
Huo et al., 2021 [18]	Chrysin	 osteogenic differentiation  proliferation  Runx-2, Col I, OCN	Posa et al., 2016 [19]Posa et al., 2018 [20]	Vitamin D	 osteogenic differentiation  Runx-2, Col I  mineral matrix deposition  α_V_β_3_ integrin expression
	Prenylflavonoids				
Nam et al., 2021 [21]	Isonymphaeol B (INB)	 mineral matrix deposition  odontogenic markers
Alipour et al., 2023 [22]	Hesperetin0.5 and 1 μM	 proliferation 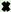 cytotoxic effect  Col I, BSP, Runx-2, OCN, ALP  mineral matrix deposition 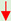 inflammation			
	Curcumin				
Samiei et al., 2021 [23]	0.5–1 μM	 proliferation 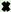 cytotoxic effect  ALP
Samiei et al., 2022 [24]	25 μM	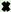 proliferation			
Son et al., 2021 [25]	Irisin	 proliferation  odontogenic differentiation  odontogenic markers  ALP  mineralized nodule formation			
	Vitamins:				
Escobar et al., 2020 [26]	Vit D/ Vit E	 osteogenic differentiation  Runx-2, OSX 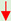 proliferation
Rasouli-Ghahroudi et al., 2017 [27]	Vitamin K2(MK-4)	 osteogenic differentiation  mineral matrix deposition			
Khanna-Jain et al., 2012 [28]	Vitamin D3	 osteogenic differentiation  OCN			
**Study**	**Non-Natural** **Compounds**	**Effects**	**Study**	**Non-Natural** **Compounds**	**Effects**
	Aspirin		Di Benedetto et al., 2020 [29]	T-LysYal	 osteogenic differentiation  Runx-2, Col I  α_V_β_3_ integrin expression
Khampatee et al., 2022 [30]	25–50 μg/mL	 odontogenesis  proliferation
Salkın et al., 2023 [13]	Ibuprofen	 proliferation			
Pang et al, 2014 [31]Galler et al., 2011 [32]Zand et al.,2023 [33]	EDTA	 osteogenic differentiation  mineral matrix deposition  proliferation  DSPP, ALP, OCN, DMP-1			

## 2. Dental Pulp Stem Cells (DPSCs)

The dental pulp (DP) is a connective tissue located in the center of the tooth and is highly vascularized. It is encircled by dentin, a mineralized hard tissue, and includes several cell types, such as undifferentiated progenitor cells and odontoblasts [34,35,36,37,38,39,40]. The DP derives from multipotent mesenchymal cells of the cranial neural crest, which migrate, during early embryonic development, towards the first and second branchial arch [37]. This dental tissue contains numerous stem cells in the perivascular niche, just like in the embryonic mesenchyme [41]. Postnatal stem cells possess immense potential for tissue regeneration and they constantly work to maintain homeostasis. The identification of these stem cells in the orofacial system occurred after the 2000s, when they were first discovered in the DP and called dental pulp stem cells (DPSCs) (Figure 1). Gronthos et al. were the first to identify and isolate DPSCs [39]. DPSCs come from the DP of permanent teeth and are usually extracted from wisdom teeth because third molars result in an extra mostly of little use for chewing and there is often not enough space for their eruption; thus, they can be extracted without critical problems and their DP tissue can be easily isolated. Grottkau et al. demonstrated the potential of these newly isolated cells that can self-renew, proliferate, and differentiate into cell lines such as osteoblasts, chondrocytes, adipocytes, neurocytes, and smooth muscle cells [40]. Recent studies show that DPSCs have the capacity, in vitro, to differentiate into bone cells and, in vivo studies, to form mineralized tissue [42]. These data suggest that DPSCs may have great value for dental and periodontal tissue engineering and efficacy on bone regeneration. Many scientists have considered these cells, thanks to their ability to differentiate into many cell types, a useful source of pluripotent stem cells to deepen the knowledge of multilineage differentiation in vitro and to adopt them for therapeutic applications.

## 3. Differentiative Role of Natural Molecules in DPSCs

Different molecules, such as vitamins, natural compounds, or drugs positively influence DPSC differentiation. The discovery of the activity of these molecules has helped researchers to know and better understand the osteogenic characteristics of these stem cells. Below we describe some molecules of natural and non-natural compounds, studied in recent years, associated with the DPSC osteogenic differentiation and activity.

### 3.1. Resveratrol

Several studies have been conducted on natural antioxidant molecules, such as resveratrol (RSV) (Figure 2a), present in grape skin, cranberries, peanuts, and root extracts of *Polygonum cuspidatum* [43]. A variety of biological functions of RSV have already been studied. It often acts as an important and functional activator of Sirtuin 1 (Sirt1) [44] and studies have been carried out on it in the field of traditional medicine [45,46].

For the first time Feng et al. [15], studied the effects of *Sirt1* in DPSCs using RSV to favor the function of *Sirt1* and siRNA to silence *Sirt1* gene expression. The silencing, regardless of the presence of RSV, led to a reduced expression of *Sirt1*. Furthermore, RSV was able to up-regulate the expression of Runx-2, BMP2, and Collagen I (Col I), while this process would have been weakened with the siRNA transfection of *Sirt1*. Interestingly the Alkaline Phosphatase (ALP) staining result confirmed RSV capacity in stimulating DPSC osteogenic differentiation. This study validates the function of RSV in promoting the osteogenic differentiation of DPSCs via Sirt1 [15].

It is well known that RSV shows antioxidant properties [45]; in this regard, Zhang et al. investigated its capacity to induce the osteogenic differentiation of DPSCs in the presence of oxidative stress [14]. Their data confirm RSV ability to potentiate osteogenesis and to be an antioxidant for DPSCs. The study shows an upregulation of Sirt1/Nrf2 expression, which appears to be reduced due to oxidative stress, and osteogenic markers such as Runx-2 and Osteocalcin (OCN) were rescued by the treatment with RSV. A similar pattern for Sirt1 expression was observed in in vivo experiments where a potentiation of the bone matrix and collagen was noted in mice that had defects in the cranial structure. These results reinforce existing data on the role of RSV as a promoter of bone formation and enhancer of DPSC proliferation; moreover, they provide useful information for regenerative medicine and for the development of bone therapies based on RSV [14].

Although RSV represents a molecule particularly suitable for bone regeneration therapies, its characteristics make it difficult to study and deepen its effects in vivo through oral administration [47]. As a matter of fact, this molecule has little solubility in water, deficient pharmacokinetics, and rapid metabolism. Therefore, to overcome these limitations, alternative administration methods are necessary to favor direct RSV action on the site to be treated.

A new strategy currently in use is represented by electrospinning membranes, with biodegradable characteristics, which can be combined with the molecule and released in a controlled manner on the affected area that presents a bone defect. There are still few works that have employed RSV with this type of membrane [48,49].

Riccitiello et al. were the first to use and study some polymers for electrospinning membranes to be used as ideal substrates to administer RSV, evaluating its effect on osteoblasts and osteoclasts [50]. They designed two materials: poly (ε-caprolactone) PCL and poly (lactic) acid (PLA). These materials have already been studied in the past both on bone marrow MSCs, where they led to an accretion of them [51], and on DPSCs that differentiated into osteoblast-like cells, overexpressing the most common markers of bone formation [28]. The authors loaded RSV on these substrates and showed that the release of the molecule on the affected area in a late manner favored the differentiation of DPSCs into osteoblast-like cells and blocked osteoclastogenesis. This study gives an interesting starting point for regenerative medicine as these materials, associated with RSV, can provide useful support to repair bone defects, and therefore help bone formation, counteracting resorption [50].

### 3.2. Flavonoids

Flavonoids are a class of organic compounds belonging to the phytonutrient family. They are natural substances found in fruits and vegetables that offer numerous health benefits [52]. Flavonoids act as antioxidants, helping fight free radicals and chronic disease. They can also mitigate inflammation, improve heart health, and reduce blood pressure [53]. Among the various studies on natural molecules, it has emerged that some flavonoids also promote osteoblast differentiation [54,55,56].

#### 3.2.1. Taxifolin

Taxifolin (dihydroquercetin) (Figure 2b) is the most common flavonoid and its presence has been demonstrated in citrons and onions, but it is also found in conifers: French maritime pine and Douglas fir [17,57]. This compound shows many beneficial biological and pharmacological properties: it displays anti-inflammatory, antioxidant, anti-apoptotic, and anti-cancer effects [58,59,60,61]. These healthful features have prompted researchers to find out whether taxifolin could protect DPSCs from apoptosis and be involved in the osteogenic differentiation process. Fu et al. revealed the surprising effects of the flavonoid on DPSCs highlighting its synergistic action with carbonic anhydrase IX (CA9), an enzyme which is often related to the aggressiveness and prognosis of many cancers. The study points out the combined effect of the two molecules in inhibiting DPSC apoptosis under inflammation and hypoxia conditions. Silencing *CA9* nullifies the effect of Taxifolin on DPSCs [17]. These data lay the bases for the possible use of anti-apoptotic and hypoxia-fighting molecules to prevent the reduction of osteogenic differentiation in inflammatory conditions. The natural molecule taxifolin could be used for bone regeneration starting from a MSCs source and could protect DPSC osteogenic differentiation in inflammatory conditions.

#### 3.2.2. Chrysin

Chrysin (5,7-dihydroxyflavone) (Figure 2c) is a natural flavonoid which can be extracted from the seeds of *Oroxylum indicum* and has wide pharmacological activity [18]. Several studies have shown that Chrysin possesses anti-inflammatory and anti-carcinogenic properties [62,63,64,65,66,67]. It is one of the polyphenolic compounds abundant in propolis, which inhibits osteoclast maturation and reduces bone resorption [68]. Studies on this flavonoid have been conducted in the past demonstrating its effect in enhancing the proliferation and osteogenic differentiation of MSCs [69,70]. An in vitro study demonstrated that Chrysin induces osteogenic differentiation and mineralization by activation of Erk1/2 signaling in the MC3T3-E1 cell line [67], but its role on DPSCs was unclear. This aspect was then investigated by Huo et al. [18]. They experimented with in vivo grafting of scaffolds with DPSCs integrated on β-tricalcium phosphate (β-TCP), treated or not with Chrysin, in nude mice and repeated the same experiment on a rat model presenting a cranial defect. It emerged that Chrysin stimulates DPSCs leading to the abundant bone formation on both models tested. Furthermore, Huo and colleagues showed that Chrysin upregulates the expression of OCN, Runx-2 and Col I proteins in DPSCs, thus demonstrating that the flavonoid enhances the osteogenic differentiation process [18]. This study could be useful in bioengineering techniques and regenerative medicine. In fact, it has indicated new patterns and new natural molecules to counteract bone defects.

#### 3.2.3. Prenylflavonoids

Prenylflavonoids are a class of compounds found in a variety of plants and known for their anti-inflammatory, antioxidant, and osteogenic effects [71]. Prenylflavonoids could be promising stimulators of osteogenic activity [72]: icariin (ICA), a compound isolated from *Epimedium pubescens* plant, and its active metabolite icaritin (ICT) have been shown to enhance differentiation and activity of rat osteoblasts in vitro. A more recent study reinforces this potential of ICT on human cells evidencing the induction of osteoblast proliferation and function [73]. The same authors later revealed an analogous effect of the prenylflavonoid on human MSCs: their proliferation, migration, and osteogenic differentiation were induced by ICT treatment [74].

A similar impact in the promotion of osteogenic activity has also been attributed to 8-prenylnarigenin, a prenylflavonoid extracted from hop, which appeared to induce the osteoblastic differentiation of rat bone marrow stromal cells (rBMSCs), through the increased expression of *Runx-2* and *Osterix* (*OSX*), with consequent enhanced mineral matrix deposition [75].

Concerning the possible effect of these natural compounds on DPSCs, several types of prenylflavonoids have been identified in the fruit of *Macaranga tanarius* and among them, nymphaeol B, isonymphaeol B (INB) (Figure 2d), and 3′-geranyl-naringenin have no cytotoxic effect on DPSCs [21]. Nam et al. identified in INB the most efficient compound in stimulating DPSC differentiation and in inducing mineralization in a dose-dependent manner. The same in vitro study proved that INB directly affects DPSC differentiation: the treatment for one week with this prenylflavonoid increased the expression of odontogenesis marker genes such as *BSP*, *matrix protein-1* (*DMP1*), and *dentin sialophosphoprotein* (*DSPP*). Likewise, after two weeks, there was also an increase of *Runx-2* expression levels. INB effect on mineralization was confirmed by in vivo experiments; the tooth bud INB treatment of post-natal day five (PN5) ICR mice significantly induced tooth root elongation when transplanted into sub-renal capsules of six-week-old mice, compared with untreated tooth buds. These data confirm that INB effectively promotes dentin formation by odontoblasts [21]. These findings suggest that prenylflavonoids may have potential applications in the field of bone bioengineering.

#### 3.2.4. Hesperetin

Citrus fruits are known to have beneficial effects on health. In particular, oranges, lemons, and grapefruits have been demonstrated to contain a flavanone called hesperetin (HS) (Figure 2e) which belongs to a subclass of flavonoids and presents antioxidant, anti-inflammatory, antimicrobial, and antiviral properties [76]. This bioflavonoid has been clinically tested, showing the capacity to promote hard tissues repair: it can be used to treat bone fractures and as a long-term preventive treatment of arthritis; HS also helps to maintain bone density and reduce the severity of joint pain [77]. Furthermore, HS has been shown to suppress the expression of the inflammation marker p65, suggesting that HS may be effective in the treatment against inflammation in preclinical and clinical models [78]. One strategy that is currently applied to make the best use of natural molecules is represented by nanotechnology. In fact, pharmacologically active molecules often have dimensions and characteristics that are not always suitable for the administration; therefore, it is possible to exploit nanotechnology to improve their bioavailability. Research has recently focused on nanomedicine, which uses nanoparticles as vectors for drug delivery, allowing treatment to be more effective than conventional methods [79]. Alipour et al. have taken advantage of this kind of approach evaluating the osteoinductive ability of HS on DPSCs through nano-HS particles. They created nanoparticles of HS, obtained using a spray dryer, from a solution of HS and acetone, and subsequently in sodium dodecyl sulfonate. Then, they treated DPSCs with HS administered as soluble molecules or nano-HS and observed that, in both treatments, DPSCs proliferated without undergoing any cytotoxic effect. In addition, when low concentrations of nano-HS (0.5 and 1 μM) were used, DPSCs showed a remarkable proliferation. Furthermore, HS nanoparticles enhanced the expression of key osteogenic markers (Col I, BSP, Runx-2, OCN and ALP), as well as mineral matrix deposition, compared to HS-treated and untreated groups [22]. As previously mentioned, similarly to other flavonoids, HS plays an anti-inflammatory role [76] and Alipour and colleagues found that HS nanoparticles reduced inflammation in DPSCs compared to the treatment with HS [22]. These studies prove that flavonoids have proliferative effects on MSCs at low concentrations and, furthermore, their osteogenic activity could be improved by the application of approaches at the nanoscale level.

### 3.3. Curcumin

Curcumin (Figure 2f) is a polyphenol found in turmeric, a well-known yellow spice widely used in Oriental cooking and traditional Chinese and Ayurvedic medicine [80] for its healing benefits [81]. It is known for its powerful anti-inflammatory, antioxidant, and anti-cancer properties [82]. Additionally, curcumin may be helpful in preventing bone loss due to osteoporosis [83]. It has been shown that curcumin stimulates osteoblastic proliferation and differentiation [84]; moreover, Samiei et al. showed that the use of curcumin and calcitriol enhanced DPSC differentiation and mineralization [23]. This study demonstrated that low doses of curcumin (0.5–1 μM) have no toxic effects and promote DPSC growth.

DPSCs respond to growth factors that promote osteogenic differentiation and mineral matrix production; in particular, curcumin stimulation increases ALP expression at both mRNA and protein levels in DPSCs [23].

Interestingly, these results indicate that curcumin could be used as a biological agent to promote osteogenic differentiation of DPSCs and could be an alternative to the drugs commonly used in clinical practice for bone regeneration. Crucial to the use of curcumin in these studies is its bioavailability, as curcumin is absorbed to a very small extent compared to the amount ingested. This means that high quantities are required to achieve the intended effects and these in turn may increase the risk of adverse effects [85]. A strategy to overcome this problem could be represented by nanotechnology. Nanoparticles complexed with curcumin could improve its bioavailability and optimize its therapeutic effects [86]. Samiei et al., in another study, used this approach and evaluated the cytotoxic effects of nanocurcumin at different concentrations on DPSCs [24]. They showed that nanocurcumin, with concentrations ranging from 0.5 to 10 μM, had no toxic effects and rather promoted DPSC proliferation. On the contrary, when used at higher concentrations (25 μM) and for extended exposure times, only the lowest doses were non-cytotoxic; furthermore, the higher concentrations inhibited DPSC growth. Nanocurcumin, therefore, could present a dose- and time-dependent effect on DPSCs and, in conclusion, it could be a feasible alternative to promote osteo/odontogenic differentiation of dental stem cells. Moreover, curcumin administered in nanoparticles exhibited high bioavailability, low toxicity, and strong biological activity, thus making it an excellent resource for hard tissue repair.

### 3.4. Irisin

Irisin (Figure 3) is a hormone especially produced in skeletal muscle during physical exercise. It has been studied for its potential health benefits, including its ability to increase energy expenditure, reduce fat mass, and improve metabolic health [87]. Son et al. investigated and demonstrated that myokine Irisin promoted odontogenic differentiation of DPSCs by inducing mineralized nodule formation, enhancing ALP activity, and upregulating odontogenic markers [25]. Moreover, the researchers evaluated a significant increase in DPSC migration through scratch wound tests. The results show that the healing of the scratch was significantly increased, due to the Irisin administration for 24 h, in which period cell proliferation was significantly enhanced both in the control group and in the group with Irisin. It is interesting to highlight that the study by Son et al. is the first of its kind which examines Irisin as an odontogenic potential of DPSCs, indicating that this myokine can enhance odontoblastic differentiation and mineralization in DPSCs.

### 3.5. Vitamins

The formation and preservation of bone health are processes that require an adequate supply of essential nutrients, including vitamins. Among vitamins related to bone health, Vitamins C and D are known to help the formation of new bone tissue. Vitamin C stimulates the production and correct structural formation of collagen, a major component of bone tissue extracellular matrix (ECM), while Vitamin D increases the absorption of calcium, which is also fundamental for bone formation [88,89,90]. In addition, Vitamin K and other vitamins such as Vitamin B6, B12, and folic acid have also been shown to have a beneficial effect on bone [91,92].

Several studies on MSCs reveal that the treatment with different vitamins can positively affect the osteoblastic differentiation of these stem cells [88,89,92,93]. Studies show that Vitamin D (Vit D) (Figure 4a) and Vitamin E (Vit E) (Figure 4b) play an important role in promoting DPSC differentiation [26,94,95]. An interesting in vitro study evaluated the effect of both Vit D and Vit E, individually or in a combined way, on DPSCs during their osteogenic differentiation [26]. Vit D treatment appeared to enhance the formation of calcified nodules, but this increase was minimal when the two vitamins were used simultaneously. This result shows that the use of the two vitamins together did not strengthen the prompting effect on osteoblastic differentiation in DPSCs. Furthermore, Vit D and Vit E, individually, stimulated an increase in the expression of cell differentiation genes such as *Runx-2* and *OSX*. While in a combined manner, in addition to a decrease in cell proliferation compared to single treatments, there were changes in the morphology of the cells following exposure to the two vitamins [26].

There are also studies on Vitamin K demonstrating positive effects on MSC osteogenic differentiation. In fact, Vitamin K2 (MK-4) (Figure 4c) has been shown to be associated with increased bone mineralization and greater resistance to fractures [91,92,96]. An in vitro study analyzed the effects of this vitamin on DPSCs used at different concentrations [27]. At 14 days, the positive effects of MK-4 were evident in the differentiation of DPSCs into osteoblasts with extracellular calcium deposition [27].

Interestingly, the effects of these vitamins are aimed at models for tissue engineering. The study by Khanna-Jain et al. highlighted the ability of DPSCs to proliferate and differentiate on a poly (l-lactic acid/caprolactone) scaffold [28], thanks to Vitamin D3 and dexamethasone, a powerful anti-inflammatory known as an inducer of osteogenic differentiation in DPSCs [97]. These two enhancers have been shown to have a potentiating role in DPSC osteogenesis, improving their adhesion to scaffolds and upregulating the osteogenic marker OCN. Therefore, vitamins may also have an interesting role in regenerative medicine.

## 4. Non-Natural Compounds

### 4.1. Aspirin

Acetylsalicylic acid (ASA) (Figure 5a), commonly known as aspirin, is a pain reliever and fever reducer that belongs to a class of drugs called nonsteroidal anti-inflammatory drugs (NSAIDs) [98]. Aspirin has been shown to have a positive effect on bone metabolism, including the capacity of increasing bone-mineral density and bone formation and of improving the renewal of bone marrow MSCs [99,100]. Aspirin can be also used in dentistry: it can accelerate the repair of damaged periodontal ligaments since it improves cellular function favoring the differentiation of stem cells and thus modulating the healing process of deep periodontal wounds [101,102]. However, the efficacy of aspirin on osteogenic differentiation of DPSCs needed further investigation in vivo and Yuan et al. studied its impact on bone repair in a rat skull defect model, using bovine Bio-Oss where DPSCs were seeded [103]. Bio-Oss is a natural alternative bone compound widely used in regenerative dentistry during surgical treatments for bone regeneration. It can become an integral part of the newly formed bone structure [102,104]. Yuan and colleagues demonstrated that, after seeding DPSCs on Bio-Oss, aspirin enhanced cell proliferation and DPSCs were able to cover the Bio-Oss scaffolds [102]. Aspirin effect on DPSC proliferation was recently also corroborated by Khampatee et al., who found an increased proliferation rate as well as an enhanced odontogenesis of DPSCs when a low dose of aspirin (25–50 μg/mL) was used in vitro [30]. These works represent a promising study for new therapies on DPSC-based bone regeneration.

### 4.2. Ibuprofen

Ibuprofen (Figure 5b) is a NSAID widely used in clinical practice to treat pain symptoms, mainly headaches and back pains and also inflammation, to reduce fever, alleviate flu and cold symptoms, and treat arthritis. This pharmacological molecule is known as an inhibitor of cyclooxygenase (COX), an enzyme involved in the production of prostaglandins, in their turn associated with the onset of inflammation and pain. Ibuprofen in low doses inhibits the COX-1 pathway, while in high doses it inhibits the COX-2 pathway [105]. Several studies have revealed that ibuprofen reduces cancer-related risks [106,107,108,109] and, moreover, decreases cancer stem cell stemness by acting on the COX-2 pathway [109]. Ibuprofen effect on DPSCs was investigated in a 2023 study; in particular, low (0.1 mmol/L) and high (3 mmol/L) concentrations were analyzed. Both concentrations, but the high concentration with a greater effect, appear to have a reducing effect on mitochondrial membrane depolarization, reduce DNA damage, and increase the viability and proliferative capacity of DPSCs. The improvement in the stemness characteristics of DPSCs related to the use of ibuprofen or other NSAIDs, according to the authors, could be useful in the treatment of inflammatory diseases such as osteoarthritis (OA). They speculate that in OA treatment ibuprofen may have similar effects as curcumin [13]; in fact, curcumin anti-inflammatory action has been demonstrated to inhibit the activation of COX-2 involved in OA processes [110]. However, further studies are needed to more closely examine the mechanisms of ibuprofen on DPSCs. At present, this study provides initial hints on the potential use and effects of ibuprofen on DPSCs.

### 4.3. EDTA

Ethylenediaminetetraacetic, known as EDTA (Figure 5c), is a carboxylic acid [111] used as a decalcifying agent and for removing toxins that accumulate on the surface of enamel and dentin [31]. It has been shown to promote DPSC differentiation through the release of growth factors by dentin [32] and also to contribute in keeping stem cells viable and healthy, preserving and promoting them [112]. These characteristics have fueled the interest of researchers to investigate the possible application of EDTA in the field of regenerative medicine. In the reconstruction therapy of the inner part of the tooth, it is necessary that stem cells adhere on the dentin surface in order to differentiate and proliferate into odontoblast-like cells [113]. Pang et al. showed the efficacy of EDTA in promoting the attachment of DPSCs to the dentin surface and stimulating the release from dentin of DPSC differentiating/promoting molecules [31]. The authors also demonstrated that EDTA enhanced cell adhesion capacity by increasing the expression of the adhesion protein fibronectin (FN). Furthermore, markers such as *DSPP* and *DMP-1* and mineralization were upregulated, thus demonstrating an effect of EDTA in the differentiation of DPSCs to the odontoblastic direction [31]. In a later study, Galler et al. confirmed that the addition of EDTA to DPSC cultures could influence migration, differentiation, and bone mineralization of this type of stem cells [114].

A recent study has compared the effects of natural and non-natural compounds on DPSC differentiation, showing that EDTA increases the protein expression level of DSPP, ALP, OCN and DMP-1. These data were in line with those obtained by treating the cells with the natural molecule curcumin, highlighting that some non-natural compounds, i.e., EDTA, may have similar effects to natural compounds stimulating DPSC proliferation and adhesion to dentin. EDTA could represent a valid option to enhance dental regeneration in patients [33].

## 5. Dental Bud Stem Cells (DBSCs)

The DB is the immature part of the teeth. It has been observed that it contains MSCs that can differentiate into osteoblast-like cells [115]. DBSCs (Figure 1) can give rise to all the tissues that we find in the mature tooth, such as enamel, pulp, dentin, periodontal ligament, and cementum. The DB, being an immature organ, consists of more undifferentiated cells than the DP [8]. The dimension represents another advantage since the DB is considerably larger than the pulp. Such properties induce the consideration of the DB as a convenient source of MSCs, but obtaining these postnatal stem cells requires the extraction of a tooth before it erupts. It is necessary to proceed with an early removal (age 7–12) of the third molar DB, using preferably a reliable technique called piezosurgery, which is a more conservative surgical approach. Furthermore, the patients selected for extraction will not lose any functional teeth, since they are often subjects who, due to the birth of a wisdom tooth, could face dental overcrowding [29].

DBSCs exhibit similarities to bone-marrow-derived MSCs. This has been verified by studying the expression pattern of adhesion molecules during DBSC differentiation [115]. Several studies have confirmed that cadherins promote the osteogenesis of bone precursor cells [104,116]. Indeed, undifferentiated DBSCs express adhesion receptors such as N-cadherin and cadherin-11, which change their localization during osteogenic differentiation, thus confirming their mesenchymal derivation and osteogenic capacity [115]. The same studies indicate that DBSCs can be used for bone formation: to fill bone gaps and to replace damaged parts of the bone. Further exploitation can be to regenerate or strengthen bone in case of fractures or bone diseases such as osteoporosis. These cells may also be employed in stomatognathic systems, such as the repair or replacement of damaged or lost teeth, making them an important resource for tissue engineering. The optimal reconstruction of bone tissue requires that stem cells have to be grown and differentiated on biomaterial scaffolds under osteogenic conditions and then implanted in vivo. To do that properly, cells need to interact not only with each other but also with their ECM to acquire and maintain adequate tissue architecture. It was found that by culturing DBSCs with ECM proteins such as osteopontin (OPN), FN, and vitronectin (VTN) there was an enhancement of the differentiation into osteoblast-like cells [115]. This interesting study suggests the application of DBSCs in the reconstruction of bone tissue, which could be remarkably improved by adding some ECM proteins to the biomaterial used.

## 6. Differentiative Role of Natural Molecules in DBSCs

### 6.1. Polydatin

Non-pharmacological therapies based on natural compounds have recently become of great interest to researchers. They are extensively studied for their antioxidant and anti-inflammatory effects [117,118]. Studies conducted on bone tissue have shown that the antioxidant properties of phytochemicals, present in fruits and vegetables, have positive effects on bone health [119]. Polydatin (Pol) (Figure 6), a natural glycosylated precursor of the well-known compound RSV, is a powerful stilbenoid polyphenol, which is present in nature and particularly abundant in *Polygonum Cuspidatum*. This glucoside, presenting numerous therapeutic potentials, has recently been the object of several studies due to its possible applications as a curative agent [120]. Regarding the effects of Pol on the skeletal system, studies have demonstrated that the molecule can have an anti-osteoporotic effect and, in particular, can induce osteogenic differentiation and migration of MSCs from bone marrow [121,122].

Of note, only the study carried out by Di Benedetto et al. focused on the involvement of the oligostilbene Pol in DBSC osteogenic differentiation [16]. The research in question investigated and compared how both RSV and Pol, affect DBSC differentiation and therefore could aid bone formation. They approached the study by verifying whether DBSCs were responsive to RSV and Pol treatment and found an enhanced Sirt1 expression when the molecules were used. Subsequently, they demonstrated that a lower dose of Pol (0.1 μM) stimulated osteogenic differentiation of DBSCs, upregulating the expression of activating transcription factor 4 (ATF-4), Osteopontin (OPN) and ALP, typical osteoblast markers, and increased the deposition of mineral matrix. On the contrary, a higher dose of Pol (1.0 μM) appeared to be ineffective on the osteogenic differentiation process. Interestingly, low doses of Pol showed an effect comparable with that of RSV used at higher concentrations [16]. These results propose an effective role of Pol in enhancing the osteogenic differentiation of DBSCs by sharing similar properties with RSV and, surprisingly, showing a real effect at considerably lower concentrations. These findings suggest that Pol may be an alternative to RSV or even more effective, strengthening its positive aspects as an antioxidant or anti-inflammatory and its strong bioactivity [123,124]. Thus, Pol could represent a valid alternative to RSV for medical or industrial applications.

### 6.2. Vitamin D

We have previously illustrated how extensively the role of vitamins in DPSC osteogenic differentiation has been investigated. On the contrary, there are few data in literature related to vitamins and their possible influence on DBSCs. In 2016, Posa et al. focused, for the first time, on the potential effect of Vit D (Figure 4a) on DBSCs differentiating into osteoblasts [19]. The experiments demonstrated that cultures of DBSCs, with a suitable osteogenic medium, showed an improvement in the differentiation into osteoblastic cells when treated with the vitamin. In fact, Vit D increased the expression of Runx-2 and Col I, typical early markers of osteoblastogenesis, and favored the formation of mineralized matrix nodules. Later, the authors investigated Vit D influence on the expression of α_V_β_3_ integrin [20], which is well known for its involvement in the commitment of MSCs towards the osteogenic lineage. The observed findings clarified that Vit D acted on the integrin expression favoring its disposition at the focal adhesion sites level. This action would drive the cells toward their differentiation and would also be supported by the presence of FN, an adhesive glycoprotein of the ECM [20].

These findings highlight that Vit D enhances the osteoblastic characteristics of DBSCs and that the supplementation of this vitamin could potentially be used to improve DBSC osteogenic capacity and therefore lead to bone regeneration applications.

### 6.3. Irisin

Irisin (Figure 3), a myokine recently identified, stimulates migration, growth, osteogenic differentiation of periodontal ligament cells (PDLCs), and odontogenic differentiation of DPSCs [25,125]. An important role of irisin has been demonstrated in bone remodeling and in fracture healing [126,127,128,129]. In a recent study, it has been observed that Irisin, bound to an αv integrin receptor, acts on osteocytes [130]. An in vitro study, conducted on the MLO-Y4 osteocyte cell line, showed that the integrin receptor, when bound by its ligand, activates one of the main signaling pathways: the phosphorylation of Erk1/2 [131]. Recently, the effect of irisin has been studied in a model of osteoblastogenesis, represented by DBSCs, focusing on the expression of the most important markers of osteoblast differentiation [132]. In this work by Posa et al., the responsiveness of DBSCs to irisin was evaluated for the first time and a cellular response emerged through Erk1/2 phosphorylation. Moreover, the cells maintained the response to the myokine and were induced to osteoblastic differentiation even when cultured under osteogenic conditions. Furthermore, there was an increase in the mineral matrix deposited by DBSCs, again under osteogenic conditions and treatment with irisin. DBSCs, by means of the myokine effect, also presented an important expression of OCN, a late marker of osteoblastic differentiation. These effects indicate that irisin plays a relevant role in the osteogenic differentiation process of DBSCs [132].

## 7. T-LysYal: A Non-Natural Compound That Promotes Osteogenic Differentiation

Other studies have investigated novel ways to enhance osteogenic differentiation and promote mineral matrix deposition [115,133,134]. Glycosaminoglycans (GAGs) are polysaccharides that play a vital role in the function of many tissues, such as skin, mucous membranes, ligaments, and tendons. They are involved in the regulation of cell permeability, cell adhesion, and tissue regeneration. GAGs are also involved in the synthesis of chemical signaling molecules, an example being hyaluronic acid (HA). HA is a polysaccharide acid abundantly present in the ECM of many tissues. Recently, a new derivative of HA has been developed, called T-LysYal (T-Lys) (Figure 7) and formed by a combination of HA, lysine hyaluronate, sodium chloride, and thymine. Research has proven encouraging results of T-Lys in tissue regeneration, particularly in bone and cartilage [29]. T-Lys has been shown to promote wound healing in the nasal mucosa [135] and the cornea [136], provide anti-inflammatory effects, and reduce the formation of scar tissue, demonstrating that it is effective in the treatment of ulcers and chronic wounds [137]. In 2020, di Benedetto et al. conducted a work on T-Lys to evaluate its effect on MSCs from DB and to study the regenerative potential of this molecule on bone [29]. They found that DBSCs responded to treatment with T-Lys, which considerably promotes the expression of Runx-2 and Col I, both at the protein and mRNA level, demonstrating its ability in enhancing the osteogenic differentiation potential of these stem cells. Furthermore, the expression of α_V_β_3_ integrin in DBSCs by administering T-Lys was examined. The literature shows that cell adhesions are crucial for differentiation processes [138,139] and that α_V_β_3_ integrin is essential for MSC commitment into the osteoblast line [115]. The research of Di Benedetto et al. revealed that T-Lys treatment increased α_V_β_3_ integrin expression directing its localization mainly at focal adhesion sites [29]. These data suggest that T-Lys influences integrin disposition by affecting DBSC differentiation.

## 8. Perspectives and Limitations

The current need in the field of bone regeneration is to develop new models to treat bone defects. The studies mentioned in this review indicate DPSCs and DBSCs as innovative and alternative postnatal stem cells sharing similar properties to bone marrow MSCs and easily accessible in sites and organs that are not essential. In fact, it has been demonstrated that these DSCs have a remarkable ability to proliferate, even if they are not inexhaustible like immortalized cell lines. A limitation is that of paying attention and avoiding too many passages during experimentation to prevent consequences that could lead to cell aging. In the field of bone regeneration and therefore to reconstruct bone tissue optimally, in addition to having ideal cell lines as MSCs, it is essential to develop adequate strategies. In recent years, researchers have developed biomaterial scaffolds with specific chemical and physical properties to support the growth and differentiation of stem cells, in osteogenic conditions, to implant in vivo. There are cases where problems have been identified with these strategies. Complications have usually emerged from the non-recruitment and non-adherence of stem cells to the scaffold. In this case, one of the major problems is the inability to obtain a sufficient cell population to ensure quality tissue regeneration. If the cells do not integrate with the scaffold, they will not be able to proliferate and differentiate effectively, limiting regeneration. Furthermore, if the cells do not remain adherent to the scaffold, they can release toxic substances that can damage surrounding stem cells.

The major limitation of this manuscript is that it is not a systematic review, but only a description of the articles presented. Furthermore, no statistical analyzes were conducted.

The works here analyzed confirm the potential of DPSCs and DBSCs to differentiate into osteoblast-like cells and, above all, highlight the role that natural and non-natural molecules could have in improving this process, thus laying the foundations for the development of future studies aimed at therapeutic applications, such as bone regeneration, based on the combined use of these cells and compounds.

We would like to encourage further research on these promising molecules, which could clarify the mechanisms underlying their action and their effects on bone health, favoring in vivo applications.

## 9. Conclusions

Studies on molecules capable of exerting beneficial effects on bone cells, or on their precursors, are of great significance for potential applications in the biological and clinical fields, regardless of whether they are natural or non-natural molecules. Although the use of natural compounds can present indisputable advantages, confirmed by many studies on them in the literature, synthetic products often allow the overcoming of limitations associated with natural molecules by opening up alternative and useful approaches.

The research works summarized in this review provide evidence that substances found in common foods, in particular fruit and vegetables, or inedible plants, in addition to manifest antioxidant, anti-inflammatory and anticancer capacities, can induce the proliferation and differentiation of d-DSCs. Similar effects have been observed for a recently discovered myokine, irisin, that is an endogenous molecule, but also for synthetic molecules. In this regard, T-Lys, for example, exploits the characteristics of the natural molecule of hyaluronic acid, acquiring better properties thanks to the chemical groups introduced through the synthesis process. Widely used synthetic products (i.e., NSAID or EDTA) have also been studied in order to evaluate their effect on d-DSCs disclosing new, easily accessible opportunities in the view of the final goal of bone reconstruction.

However, there is a growing need to perform more in vivo studies and clinical trials to further understand the mechanisms of action and, moreover, establish the efficacy of the analyzed molecules on d-DSCs, which could represent a powerful therapeutic tool and lead to new approaches useful to repair and regenerate damaged tissue in the regenerative medicine field.

## Figures and Tables

**Figure 1 ijms-24-09897-f001:**
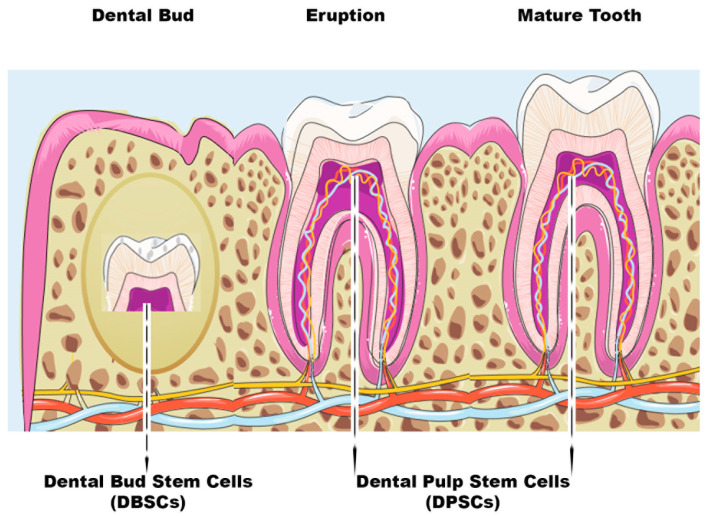
Scheme of dental bud stem cells (DBSCs) and dental pulp stem cells (DPSCs). The figure summarizes the stages of tooth growth. As discussed in the review, DBSCs can be obtained from the dental bud and DPSCs can be isolated both at the eruption or mature stage of the tooth.

**Figure 2 ijms-24-09897-f002:**
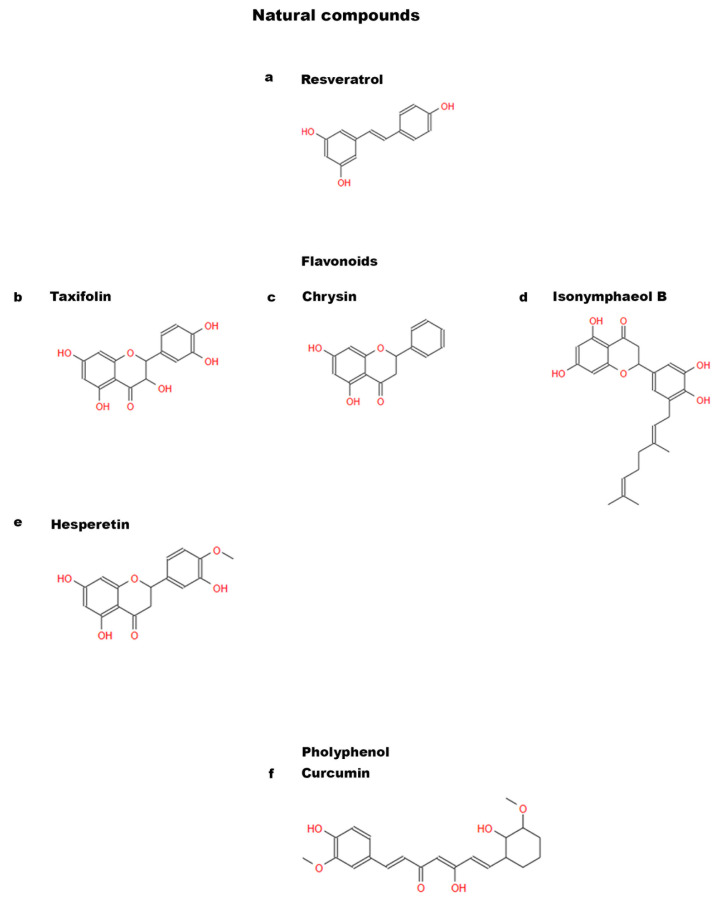
Chemical structures of natural compounds. (**a**) Resveratrol; (**b**) Taxifolin; (**c**) Chrysin; (**d**) Isonymphaeol B; (**e**) Hesperetin; and (**f**) Curcumin.

**Figure 3 ijms-24-09897-f003:**
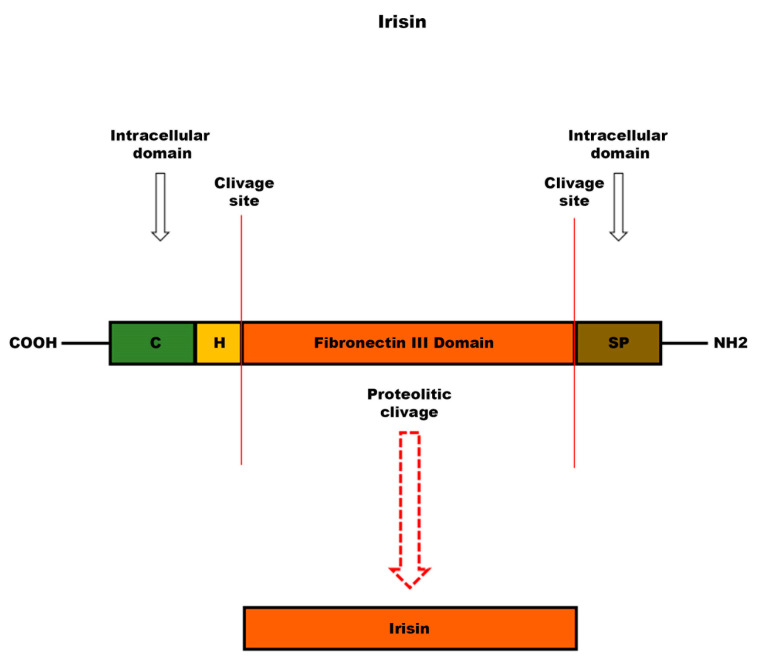
Schematic structure of irisin precursor: fibronectin type III domain-containing protein 5 (FNDC5). Following cleavage of FNDC5 at the level of the extracellular domain, irisin, a peptide consisting of a 112 amino acid sequence, is released into the bloodstream. The abbreviations indicate: C, cytoplasmic domain; H, hydrophobic domain; SP, signal peptide.

**Figure 4 ijms-24-09897-f004:**
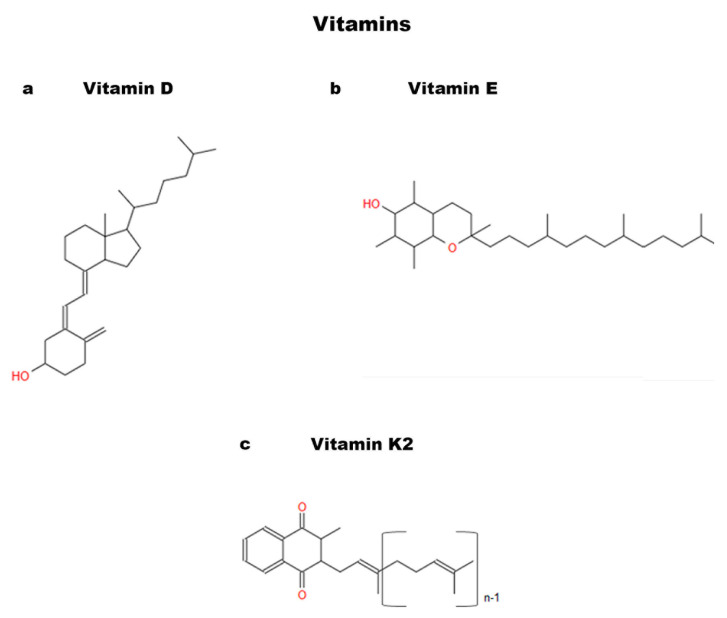
Chemical structures of Vitamins: (**a**) Vitamin D; (**b**) Vitamin E; and (**c**) Vitamin K2.

**Figure 5 ijms-24-09897-f005:**
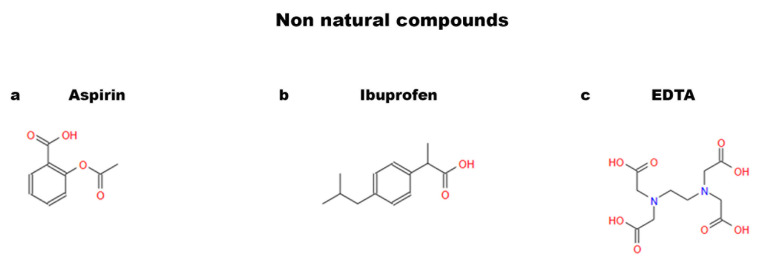
Chemical structures of non-natural compounds: (**a**) aspirin; (**b**) ibuprofen; and (**c**) EDTA.

**Figure 6 ijms-24-09897-f006:**
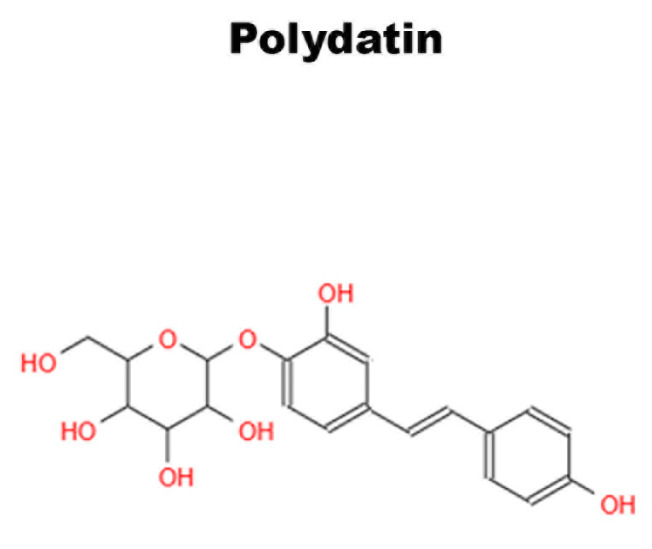
Chemical structure of Polydatin.

**Figure 7 ijms-24-09897-f007:**
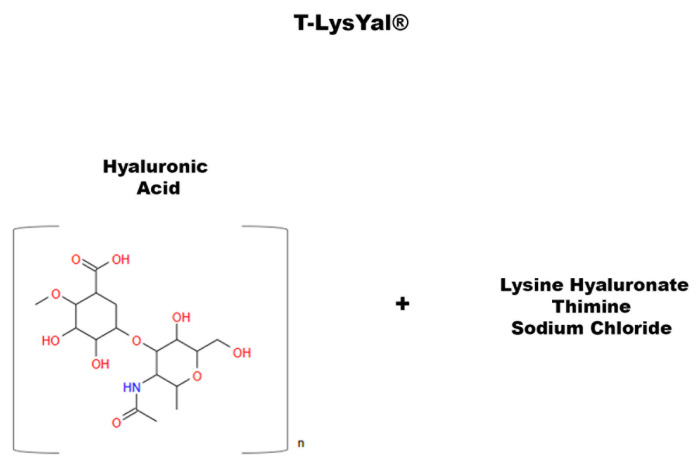
Chemical structure of T-LysYal.

## Data Availability

Data sharing not applicable. No new data were created or analyzed in this study. Data sharing is not applicable to this article.

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
