# Peer review of "Molecules Inducing Dental Stem Cells Differentiation and Bone Regeneration: State of the Art"

_ijms, 2023, doi:10.3390/ijms24129897_

Round 1
Reviewer 1 Report
This review analyses the potential and influencing factors of dental pulp stem cells (DPSCs) and dental bud stem cells (DBSCs) in dental tissue for the reconstruction of bone defects. This has certain guiding significance for the bioengineering of bone. However, it needs to be further revised to meet the publishing requirements. The frame of the article is messy.
1. In 3.2.2 “Prenylflavonoids”, Supplementation of relevant information regarding studies on the effects of prenylflavonoids on bone cells should not be limited to the effects on DPSCs.
2. The biomaterial scaffolds with specific chemical and physical properties can be introduced in a separate chapter.
3. Pay attention to the format of the article, like line 227, 230, 290, 419…
4. Adjust the framework of the article, focusing on DBSCs and DPSCs, and describe the influence of natural and unnatural compounds on it respectively.
5. Additional relevant information about the research on the influence of unnatural molecular compounds on induced proliferation and osteogenic differentiation of the two d-DSC populations.
6. need to add more Figures and Tables to make the article more readable
need to improve the English Language
Reviewer 2 Report
Dear authors,
In this review, you sought to investigate two different types of mesenchymal stem cells in relation to tissue engineering. Overall, the paper reads well and the work has been presented in a decent and systematic manner. Please find my comments below.
You will need to justify why a conclusion section is lacking from the present form of the paper as I do not believe that omitting such a section is in line with the Journal’s policy. Likewise, organization of the manuscript appears to be slightly problematic as there is no finding/conclusion of the abstract.
Another major comment of mine is related to the level of evidence and the way you present your findings in the text. To be more precise you will need to add some modesty in your statements in multiple sentences in the paper, as you are only backing up your presentation with studies of low quality of evidence. For example, in line 215, I would advise you revise your strong statement that findings are ‘confirmed’.
Also, some further important limitations need to be mentioned in the ‘Perspective and limitations’ section of your article. In more detail, I suggest you comment on the fact that yours is not a systematic review and as such the literature was not systematically searched for papers. On top of that you did not conduct any type of statistical analysis which I agree with given the nature of the included papers. However the fact that you only described other authors’ work in a qualitative manner needs to be mentioned here as well.
Please find some more specific comments below.
Line 17: Why reviewing only research work published over the last 7 years?
Line 27: The structure of the introduction section needs to be revised and I would advise you create two separate paragraphs to facilitate reading.
Line 48: Give reference.
Line 136: Again, structural organization of this section appears problematic and I would advise you merge the first paragraph with the remainder of the section.
Line 263: As a Medical Doctor, I believe that this statement is misleading/inaccurate and therefore I would advise you consider revising.
Line 270: Please revise the words ‘Efficacious effect’. Consider replacing the word efficacious with beneficial.
Slight English Language editing is needed.
Reviewer 3 Report
The authors have presented a review on new molecules that may soon be viable options for bone regrow specifically using dental stem cells. The authors review has a significant focus on natural materials, and gives two examples of non-natural compounds.
Overall the review was interesting and informative. There were a few question and/or comments that the authors should consider:
(1) the title indicates "New Discoveries..." but it was not clear what was considered new. The reader could look through the reference and make some assumptions, but a clear statement on what dates were researched for the article or what would be considered new. Bone regeneration has been an active area of research for several decades so it was not clear. The table looks like it addresses papers from 2016 on, but one sentence would clarify for all readers.
(2) The table is an excellent way to organize all the molecules. Consider referencing earlier in the manuscript. As the reader I was hoping for a way to compare and only saw the table after reading the perspectives and limitations.
(3) There was a STRONG emphasis on natural products as differentiation inducers. The authors may want to rethink the title as natural molecules. As a reader, there is a clear bias in one direction and there are many, many non-natural products that have been used - even in the last several years. You can just reference non-natural products in a short paragraph
(4) If the authors would like a more broad review of the field, there MUST be more non-natural molecule included. If the message the authors are trying to convey to the readers is the active work in the field, then all work (good and bad) should be included. The authors can draw from perspective from the data presented as long as it not biased.
The manuscript was well written and only minor grammar mistakes should be addressed.
Reviewer 4 Report
The review by Ariano et al. discussed a very interesting topical about the new discoveries on molecules inducing dental stem cells differentiation and bone regeneration. The authors described a comprehensive description of the development of differentiative role of molecules in Dental Pulp Stem Cells (DPSCs) and Dental Bud Stem Cells (DBSCs). The introduction is clearly presented to point out the importance. The section is nicely defined and is demonstrated in details. Overall, it is an informative review which warrant publishing. I would recommend the manuscript for publication after following revisions
1. Line 58: should write “Dental Pulp Stem Cells (DPSCs)” rather than DPSCs ony
2. Line 138: Should write “3.2.1 Taxifolin”. Numbering should start from taxifolin rather than Chrysin.
3. Should provide the source of taxifolin.
4. Line 136-137: should provide more details about flavonoids.
5. Line 318: should write “Dental Bud Stem Cells (DBSCs)” rather than DBSCs only.
6. Author are encourage to provide the figure containing the structure of Resveratrol, Taxifolin, Chrysin, Prenylflavonoids, Hesperetin, Curcumin and Aspirin.
7. The format of reference in not same in the manuscript for example reference 15, 21, 33, 34, 38. The reference format need to be checked carefully.
Minor editing is required
Round 2
Reviewer 1 Report
it is better edited
no suggestion
Reviewer 2 Report
Dear authors,
Thank you for submitting your revised manuscript. I have no further comments and I believe the paper basically meets the publication requirements.
Reviewer 3 Report
The authors have made several excellent revisions and significantly improved the manuscript. Given all the changes, additional of images, and structural changes. The paper should be accepted as revised.
Major English issue have been resolved in the revisions. Minor issues could be addressed during final edits with the journal office.
Reviewer 4 Report
Accept in Present form
Minor editing is required